# Poodle: Seamlessly Scaling Down Large Language Models with Just-in-Time Model Replacement

Nils Strassenburg
nils.strassenburg@hpi.de
Hasso Plattner Institute, Uni Potsdam

Boris Glavic
bglavic@uic.edu
University of Illinois Chicago

Tilmann Rabl
tilmann.rabl@hpi.de
Hasso Plattner Institute, Uni Potsdam

## ABSTRACT

Businesses increasingly rely on large language models (LLMs) to automate simple repetitive tasks instead of developing custom machine learning models. LLMs require few, if any, training examples and can be utilized by users without expertise in model development. However, this comes at the cost of substantially higher resource and energy consumption compared to smaller models, which often achieve similar predictive performance for simple tasks.

In this paper, we present our vision for *just-in-time model replacement (JITR)*, where, upon identifying a recurring task in calls to an LLM, the model is replaced transparently with a cheaper alternative that performs well for this specific task. JITR retains the ease of use and low development effort of LLMs, while saving significant cost and energy. We discuss the main challenges in realizing our vision regarding the identification of recurring tasks and the creation of a custom model. Specifically, we argue that *model search* and transfer learning will play a crucial role in JITR to efficiently identify and fine-tune models for a recurring task. Using our JITR prototype *Poodle*, we achieve significant savings for exemplary tasks.

**VLDB Workshop Reference Format:**
Nils Strassenburg, Boris Glavic, and Tilmann Rabl. Poodle: Seamlessly Scaling Down Large Language Models with Just-in-Time Model Replacement. VLDB 2026 Workshop: Novel Optimizations for Visionary AI Systems (NOVAS).

**VLDB Workshop Artifact Availability:**
The source code, data, and/or other artifacts have been made available at https://github.com/hpides/poodle.

## 1 INTRODUCTION

With the advent of large language models (LLMs) and their availability through cloud-based services, using machine learning (ML) has never been so easy. LLM providers offer simple API integration, zero model development cost for customers, no need for extensive data collection or labeling, and instant access to state-of-the-art capabilities without requiring AI expertise. Organizations that previously developed custom models now use LLMs to reduce development cost and time, while organizations without prior ML experience adopt LLMs for ML automation. As a result, companies such as Snowflake, Google, or McKinsey report that many organizations offload simple, recurring tasks such as user sentiment classification,

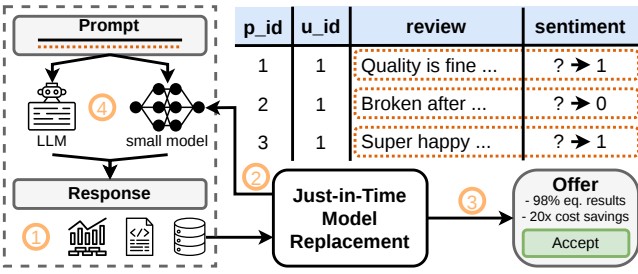

**Figure 1: Sentiment classification use case: (1) detect recurring task, (2) develop and monitor surrogate model, (3) ask user to switch, and (4) replace LLM with surrogate model.**

satisfaction classification, or churn risk identification to complex LLMs [9, 10, 15, 23, 26] even though they could be handled by much smaller, specialized models at lower cost. Furthermore, the increasing use of semantic predicates [21] in analytical workloads results in more repetitive and predictable LLMs calls.

As a running example, consider the following scenario shown in Figure 1: Startup S continuously collects product reviews in a database table. To monitor customer satisfaction, S uses a flagship LLM to predict sentiment from each review's text. If the company has access to a database with semantic operators, this can be expressed in SQL. Otherwise, a developer is tasked with prompt engineering and writing the glue code to process each review by calling the LLM provider's API, parsing the response, and inserting it into the sentiment column. In both cases, an LLM is typically chosen even though a smaller model would suffice for review sentiment classification. However, developing this model results in a longer development cycle and requires significantly more expertise and computational resources to match the LLM's accuracy. The developer would have to (i) generate a training dataset, (ii) find an appropriate model (e.g., on *HuggingFace* [19]), and fine-tune it, (iii) test and deploy it, and (iv) monitor its performance. Thus, even though an LLM results in higher operational cost, the company still opts for this option due to significantly lower development cost.

In this paper, we present *just-in-time model replacement (JITR)*, our vision for combining the low development effort of LLMs with the lower inference latency and resource requirements of specialized models. Similar to *just-in-time compilation*, where interpreted code is replaced with compiled code, a JITR system replaces LLMs with surrogate models in four steps as shown in Figure 1. ① The JITR system monitors LLM requests, identifies *recurring tasks*, and records request-response pairs. ② Once sufficient evidence has been observed for a recurring task, a *surrogate model* is generated using the recorded responses as training data. ③ If the surrogate model performs well and the user approves, ④ future requests are handled using the surrogate model, whose performance is monitored to intervene if necessary.

We envision JITR being applied in many different scenarios. First, LLM providers can leverage JITR to lower serving costs and offer new pricing models. For example, upon detecting repeated instances of S's sentiment analysis task, a provider could offer S a price reduction for switching to a fine-tuned *BERT* model. Second, organizations deploying LLMs on-premises may use JITR to reduce cost and energy consumption for recurring tasks. Third, LLM-augmented database engines [10, 20, 23, 24, 33] that answer queries using semantic operators over structured and unstructured data can benefit from JITR. With JITR, we extend the pool of models these systems can choose from by significantly cheaper expert models. For cascading-based approaches as used in LOTUS [33] and Snowflake Cortex [23], JITR enables replacing the proxy with an order-of-magnitude cheaper expert model or extending the cascade with a cost-efficient model. For approaches that explore the Pareto front, such as Palimpzest [24], JITR introduces an additional, highly cost-efficient model option that pushes the frontier further. For approaches that pair an embedding model with a cheap proxy [10], JITR provides a task-specific cheaper alternative.

JITR is not the first proposal to reduce inference cost by replacing LLMs with cheaper models, but it is unique in being specifically designed to automatically detect and optimize recurring tasks. Qin et al. [34] and Cai et al. [7] propose to automatically generate Python code for a task using an LLM. While this works for non-semantic operations, such as extracting text from REST API responses, it fails for tasks like sentiment classification. Given a task, Shen et al. [38] use an LLM to coordinate task solving by finding and combining different models on HuggingFace. Models are selected solely based on their description. However, even high-quality documentation is often not sufficient for predicting the performance of a model on a new task [4, 22, 36, 40]. Routing approaches [30] route requests among a fixed set of models, but do not extend the model pool with specialized models. Inference optimization techniques, such as quantization, are complementary to JITR.

For JITR to be effective, several conditions have to be met: (i) recurring tasks can be identified successfully from requests to an LLM; (ii) the logged requests and LLM responses provide sufficient training data for developing a surrogate model; (iii) the generated surrogate model has satisfactory performance; (iv) monitoring can detect performance degradation of deployed surrogate models and intervene; and (v) the total cost of task identification, model development, and monitoring can be amortized due to the reduced inference cost of the surrogate model.

We identify condition (v) as the most challenging due to the high cost of developing a surrogate model. We propose to generate a surrogate model by using the training data generated during task identification to first automatically select a pre-trained base model, and afterward perform knowledge distillation [17]. The rationale for extending model development beyond traditional knowledge distillation is that, when the right base model is chosen, fine-tuning requires significantly less training data, converges faster, and achieves higher accuracy than training a new model from scratch [36, 37]. This increases the likelihood that the resources spent on identifying the task and creating the surrogate model will be amortized by repeated execution of the task with the surrogate model. Thus, the key to minimizing surrogate model development overheads and achieving our vision of JITR is to develop a performant model store

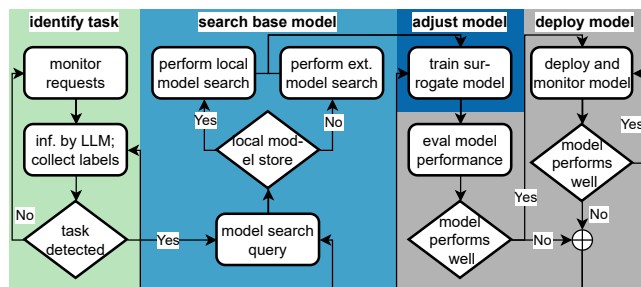

**Figure 2: The just-in-time model replacement workflow.**

offering effective techniques for searching based models for fine-tuning. While existing model search approaches have been shown to be effective [22, 36, 37], further research is needed to speed up model search from a systems perspective and to scale model search to a rapidly increasing number of available base models [19]. The main contributions of this work are:

- We present our vision for *JITR* to reduce inference cost by transparently replacing LLMs with surrogate models, discuss challenges, and identify research questions (Section 2).
- We introduce *Poodle*, a prototype JITR system built on top of an efficient model search system (Section 3).
- We conduct a preliminary evaluation of the trade-offs involved in JITR (Section 4).

## 2 JUST-IN-TIME MODEL REPLACEMENT

In this section, we discuss challenges, solutions, and open research questions in developing a just-in-time model replacement (JITR) system, covering all steps shown in Figure 2. We focus on an efficient model store supporting JITR through fast model search since it represents the most critical bottleneck, and the area where the data management community is uniquely positioned to contribute.

We envision several alternatives of user interaction ranging from (i) *no user involvement:* the system switches to surrogate models without informing or involving the user to (ii) *full user control:* the user has to explicitly approve the use of a surrogate model and what action is taken when a surrogate model exhibits performance degradation. A JITR framework may be employed either by the LLM host or by the client transparent to the LLM host.

### 2.1 Task Identification and Dataset Generation

As shown in Figure 2, assuming an LLM is currently used to handle a recurring task, the first step in developing a surrogate model is to identify the recurring task by monitoring and saving incoming requests and responses. A recurring task can be represented as a *template prompt* for the LLM, which is *instantiated* into a concrete prompt by binding values to the template's parameters. In addition, a recurring task is associated with task metadata (e.g., input/output modalities and task type), performance metadata (e.g., request frequency, latency, and throughput), and a task dataset consisting of incoming requests paired with corresponding responses or ground-truth labels when available.

*Challenges.* The main challenge in this step is to detect the recurring task as early as possible and gather associated data and metadata without significant overhead during LLM inference.

*Possible Solutions.* Possible approaches for task detection range from user-provided task information to fully automatic task detection. The easiest approach from a systems perspective is to let the user provide labeled data examples, a prompt template, the task metadata, and minimum performance requirements in a structured format. Lightweight approaches include extracting a task template for recurring tasks from semantic SQL operators, extend common LLM application frameworks such as LangChain [8] to allow users to specify recurring task, or to expect users to describe recurring tasks in natural language from which we extract all relevant task information. To automatically detect a recurring task, we can continuously monitor user inputs and analyze their plain text, task embeddings, or the LLM inference engine's KV cache. A more explicit way to detect a recurring task and extract metadata is to wrap a subset of incoming requests in an additional prompt that instructs the LLM to simultaneously answer the request and extract task information. We can do this online to optimize for costs or offline to optimize for latency. These methods can also be combined.

*Open Research Questions.* A key research question is how to balance task metadata quality against detection overhead. The higher the metadata quality, the better our decisions on the model development-versus-inference resource-savings trade-off. For example, knowing up front how often a task will recur and what the performance requirements are enables strategic resource allocation. Obtaining high-quality metadata requires either greater user effort or a more elaborate detection technique, resulting in higher costs or latency. Another research question is when a sufficient number of requests has been observed to start surrogate model development. Earlier triggering increases inference cost reduction by replacing the LLM sooner, but raises the risk of deploying a surrogate model with insufficient accuracy, wasting development resources.

## 2.2 Base Model Search

Once a recurring task is detected, JITR replaces the LLM with a surrogate model that minimizes resource consumption while meeting accuracy constraints. For JITR to be effective, we need to minimize the cost and time to monitor the requests, collect training data, and train the surrogate model, which often makes training from scratch infeasible. Instead, as shown in Figure 2, we propose to collect training data, use the data to find an appropriate base model using model search [37, 40], and optionally fine-tune this model. Models may either come from a private model store, if available, or a public model store such as HuggingFace [19].

The naive approach to model search is to let a human expert select a model from a model store, which requires expert knowledge, often leads to non-optimal choices [36], and is expensive. In response, various model search approaches have been developed [18, 22, 36]. All share the prohibitively expensive baseline of exhaustively fine-tuning all models in the model store, and approximate it in different ways [37]. Most commonly, they rank candidate models by approximating their post-fine-tuning accuracy in two steps. First, they use the models' feature extractor (all layers except the final classification layer) to transform the input data into a feature space. Next, they train a proxy model (e.g., a fully connected layer) on these features to estimate the task fit, yielding

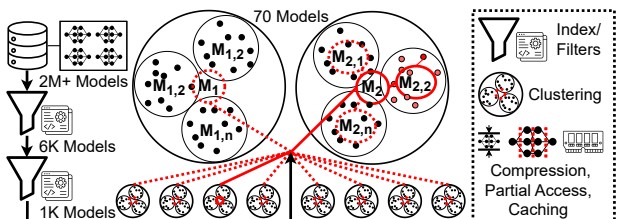

**Figure 3: Approximate hierarchical clustered model search. Dotted red lines analyzed models, solid lines selected models.**

a proxy score (the accuracy of the proxy model) as an estimate of final model performance, or apply a training-free transferability score such as LEEP [28].

*Challenges.* While approximate approaches scale better than fine-tuning all models, searching over hundreds of models based on extracted features still takes hours to days [37] and, when applied to thousands or millions of models, search can become more time-consuming than developing a single model [37, 40]. Thus, reducing the time for model search remains the greatest challenge for JITR.

*Possible Solutions.* The most expensive step in feature-based model search is feature extraction (i.e., loading the model to the GPU and performing inference). There are two main directions for scaling model search, both requiring a co-designed model store: (i) minimize the number of models considered for feature-based search, and (ii) reduce model loading and inference times.

To minimize the search space, we filter models by architecture type, inference latency, memory consumption, and benchmark accuracy on previously seen datasets. We also hierarchically cluster models on, for example, task-to-vec embeddings, intermediate inference results, and other metadata [4, 31], allowing us to evaluate models in multiple rounds: starting with cluster representatives, pruning non-promising ones, and exploring promising clusters further. This requires the underlying model store to index models for advanced filtering and support efficient hierarchical clustering.

To reduce feature extraction time, we prune models on a subset of the data [37], maximize caching of partial models and intermediate results [27, 40, 41], and load approximated model versions (e.g., quantized or pruned) in early exploration stages to identify the most promising clusters, progressively increasing precision in later stages. A model store should therefore enable block-wise model access and parameter deduplication as well as fast model access via compressed or approximate model formats [27, 43].

Figure 3 illustrates the search procedure described above. Starting from a model pool of over 2M models, we filter to a set of 1000 models across multiple stages and cluster them into three levels of 10, 100, and 1000 models, respectively. At level 0, we evaluate ten models $(M_1, \ldots, M_{10})$ and select the two clusters associated with $M_2$ and $M_5$ as the most promising ones. On level 1, we evaluate ten models per selected cluster (e.g., $M_{2,1}, \ldots, M_{2,10}$) and again select the two most promising clusters, leading to an evaluation of 40 models at level 2, which totals 70 models for the entire search.

*Open Research Questions – Metadata.* A key research direction is to investigate which metadata types are most valuable for model search and what trade-offs arise in deciding which to store. Models

can be annotated with a variety of metadata, such as training data or statistics thereof, the base model, data preprocessing pipeline details, input shape, task type, loss curves, number of parameters, model size, inference latency across hardware platforms, and accuracy on benchmarking datasets [31]. However, not all metadata is effective for model search, and metadata types differ in the effort required to extract and store them [42, 45]. For example, the number of model parameters can be determined automatically to effectively filter large collections of models. Storing intermediate results produced during model inference on reference datasets helps with clustering and speeds up model search, but comes at a high storage cost. Inference latency on unseen hardware is difficult to estimate, yet crucial for model selection. Finally, it is also important to evaluate how to efficiently store and index all metadata so that it can be rapidly accessed during model search. Ultimately, we expect a mix of metadata to be most effective. Lightweight metadata, such as input shape and task type, are easy to extract, small in size, and static, which makes them well-suited for pre-filtering. Richer metadata, such as intermediate inference results, are more discriminative but expensive to extract and store; limiting it to a representative subset of models can bound both cost and storage.

*Open Research Questions – Model Indexing and Clustering.* Another research direction is model indexing and clustering [31]. Open questions include identifying which features are most effective for measuring model similarity and determining which clustering algorithms perform best for organizing models. A key challenge is to assess whether a single, universal clustering scheme can serve all model search queries, or whether multiple specialized clusters or indices are required. A resulting research question is how to evaluate cluster suitability. Potential approaches include: selecting a representative model for testing, generating a new model to approximate the entire cluster, or to try and estimate the range of possible predictions for multiple models at once [27].

*Open Research Questions – Fast Model Access.* Performing inference requires loading each model into memory and can become a major bottleneck [40]. Compact storage formats and accelerating model loading, therefore, represent an important research direction. Key questions include whether searching over approximated models results in the same model ranking or similar proxy scores compared to those from full-precision models and how storage formats can be optimized for faster loading. Exploring storage designs that are aware of hardware and cloud-specific characteristics could substantially improve model retrieval efficiency.

*Open Research Questions – Multi Tenancy.* Approaches like Alsatian [40] show that deduplicating computations and model loading within a single model search query yields significant speedups. At the scale of modern LLM providers, multi-query optimization to detect shared computations or overlapping data access patterns across queries is a promising direction to improve system efficiency and throughput. Another direction involves tracking past search results and success rates. For new requests, this enables prioritizing models that have demonstrated strong performance on similar tasks and pruning historically underperforming models early.

## 2.3 Surrogate Model Development

Once model search identifies a promising base model, we evaluate whether it performs well enough without further adjustments, or adapt it for the specific recurring task. The main challenge is to minimize adaptation time and the time until surrogate model deployment. The most promising approach is to fine-tune the model returned from model search using the labeled data generated by the LLM, distilling the knowledge [17] from the LLM into the surrogate model.

*Open Research Questions.* Several open questions remain regarding the optimal model development strategy. First, we need to determine what types of knowledge distillation to employ and whether we should extract more than just the final predictions from the LLM, which would add overhead during LLM inference. Second, it is unclear how to predict how much training data is needed to achieve robust model performance. Third, we must investigate to what extent fine-tuning or distillation can compensate for suboptimal choices made during model search. Finally, there is the question of whether we should apply additional model compression techniques beyond distillation to further improve efficiency.

## 2.4 Model Evaluation and Monitoring

Once we have generated a surrogate model, the question is whether or not to replace the LLM. As shown in Figure 2, the replacement process involves multiple stages: identifying whether the surrogate model is a suitable candidate, deploying it if appropriate, and continuously monitoring its performance to ensure consistent quality. If the surrogate model proves unsuitable or its performance degrades during deployment, we must adjust our approach by returning to earlier stages such as collecting more labeled data, conducting model search, or performing model fine-tuning.

*Challenges.* The main challenges we face are: (i) How do we validate that a surrogate model is a good candidate to replace the LLM? (ii) Once deployed, how do we ensure consistently high performance, even in the event of distribution shift? (iii) How do we choose the right tradeoff between competing constraints on cost, inference latency, throughput, and accuracy? (iv) Which earlier stages of the JITR should be repeated when adjustments are needed?

*Possible Solutions.* A possible solution to challenge (i) is to collect a validation dataset during task identification, model search, and model development to evaluate the surrogate model's accuracy, requiring only a short benchmarking period to obtain performance metrics. A different solution is to deploy the surrogate model alongside the LLM for a certain period of time to compare the surrogate model's accuracy and other performance metrics to those of the LLM. For challenge (ii), we can also take the approach of parallel deployment, regularly routing a fraction of requests to both the LLM and surrogate model to compare their performance. We can also allow the user to add additional labeled training examples at any point after surrogate model deployment. The monitor can then employ distribution shift detection [35] to determine whether the model needs to be retrained. For challenge (iii), one solution is attempting to match the LLM's accuracy within a narrow range at all costs while compromising on cost, inference latency, and throughput reduction. Over time, we can incrementally widen the accuracy threshold and focus more on performance metrics.

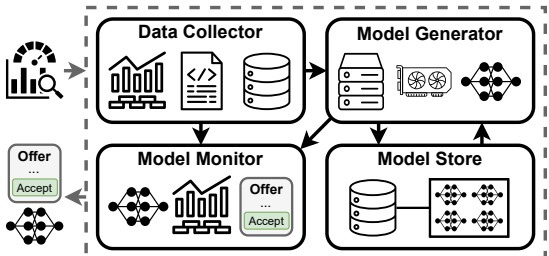

**Figure 4: Just-in-time model replacement architecture.**

*Open Research Questions.* Several open research questions remain regarding the optimal monitoring strategy and performance tradeoffs. We need to determine the right monitoring level and amount of data to collect since unnecessary monitoring introduces overhead. We also need to establish how to prioritize the different metrics: inference latency, memory consumption, throughput, and accuracy. Equally important is maintaining consistent predictive performance, since customers are sensitive to fluctuations in prediction quality, which raises the question of how frequently we can switch models without negatively impacting user experience. Similar to the trade-off for surrogate model development, decreasing the length of the trial period for a model and frequency of monitoring during deployment increases the risk of failing to detect a poorly performing model, but results in larger cost savings.

## 3 *POODLE*: A JUST-IN-TIME MODEL REPLACEMENT PROTOTYPE

In this section, we present *Poodle*, a proof-of-concept prototype for JITR. As shown in Figure 4, *Poodle* consists of four components.

The *Data Collector* continuously monitors incoming requests and the LLM's responses to collect a dataset for training. Additionally, it gathers metadata and performance metrics for request serving, including cost, inference latency, and the number of tokens. For all additional recurring task metadata, such as the input data type, the task type, and other metrics, *Poodle* groups incoming requests by their prefix and, for a subset of requests, uses an additional prompt to simultaneously extract metadata and answer the user query.

The *Model Generator* receives task definitions from the Data Collector, including the training dataset collected for the task. It uses a subset of the target dataset to issue a model search request to the *Model Store*. The Model Store follows Alsatian's [40] baseline and ranks all models by approximating how well a specific model will perform when being fully fine-tuned. The Model Generator uses the training dataset consisting of LLM requests and responses to distill [17] the knowledge of the LLM into the small model via fine-tuning to create a surrogate model. Finally, it persists the surrogate model in the Model Store for future use.

The *Model Monitor* evaluates candidate models against the deployed LLM. Utilizing and extending the data collected by the Data Collector, the Model Monitor tracks inference latency, accuracy, and costs for both models. Once a candidate demonstrates comparable accuracy at a lower cost, the monitor automatically replaces the model, optionally asking the user for approval.

After deployment, the Model Monitor continuously compares the LLM's and the new model's performance to detect data drift. While we leave the implementation of calculating energy consumption

**Table 1: Model pricing per 1M tokens, April 2026. Models marked with † show TogetherAI prices from August 2025.**

| Model | Input ($) | Output ($) | Ref |
|---|---|---|---|
| GPT-5.5 | 5.00 | 30.00 | [1] |
| GPT-5.4-mini | 0.75 | 4.50 | [1] |
| Gemini 2.5 Flash | 0.30 | 2.50 | [16] |
| Llama 3 8B† | 0.20 | 0.20 | [2] |
| BERT 80M† | 0.01 | 0.01 | [3] |

**(a) Gemini 2.5 Flash**  **(b) GPT-5.4 mini**  **(c) GPT 5.5**

**Figure 5: Cost analysis when replacing an LLM.**

or the $CO_2$ footprint, as well as continuous monitoring, for future work, we estimate the cost savings as follows. For the LLM cost, we track the number of input and output tokens and multiply them by the cost as listed by the LLM provider. To estimate *Poodle*'s cost, we sum (i) the cost to process the first $i$ wrapped requests, (ii) an estimate of the surrogate model development cost, and (iii) the cost for processing remaining requests with the surrogate model.

## 4 PRELIMINARY RESULTS

We conduct preliminary experiments to demonstrate the effectiveness of JITR. We examine cost reduction in Section 4.1, inference time reduction in Section 4.2, analyze the effect of JITR on accuracy in Section 4.3, and show that model search followed by fine-tuning outperforms other model development approaches in Section 4.4. Unless stated otherwise, we perform sentiment classification on the IMDB dataset [25] and assume that every request processed by the LLM is wrapped in an additional prompt for metadata extraction, resulting in conservative cost and latency savings estimates.

### 4.1 Monetary Cost

To determine the cost break-even point, which is the number of requests after which JITR is cheaper than using an LLM, we use the prices from Table 1, assume a switch from the LLM to a custom *BERT* model after 5k requests, and assign a development cost of $4, which roughly equals three hours of an AWS A10G GPU instance.

Figure 5 shows the break-even point and cost reduction for different models. For Gemini 2.5 Flash (used in SemBench [21]) *Poodle* breaks even after roughly 40K requests, and reduces the costs from $141 to $11 for 1M requests. With larger models such as GPT-5.4 mini, the break-even point is 16K, and for GPT 5.5, the costs break even after 8K requests, with more than $2,230 saved for 1M requests. **Takeaway:** *Under conservative assumptions for task identification, JITR amortizes its overhead after a moderate number of instances and achieves significant cost reductions at scale.*

### 4.2 Inference Time

We determine the inference time break-even point, which is the number of requests after which JITR with *Poodle* becomes faster than using an LLM. We use the IMDB dataset and evaluate different

**Table 2: Accuracy of BERT model fine-tuned on IMDB data.**

| | Ground Truth | | LLM-Generated | |
| # Items | Accuracy | Epochs | Accuracy | Epochs |
|---|---|---|---|---|
| 500 | 0.86 | 9 | 0.88 | 6 |
| 1,000 | 0.88 | 5 | 0.88 | 7 |
| 2,000 | 0.89 | 3 | 0.88 | 5 |
| 5,000 | 0.90 | 5 | 0.90 | 2 |

batch sizes on an RTX A5000 GPU for (i) a BERT model, (ii) a Llama-2-7B model using a base prompt, and (iii) a Llama-2-7B model using a longer (wrapped) prompt. An LLM approach using Llama-2-7B and a JITR approach (starting with Llama-2-7B and switching to BERT after 5,000 requests) break even within the first 100K requests. Llama-2-7B processes 13 items per second at a maximum batch size of 16 while BERT processes 19.6× more items in the same time at a maximum batch size of 128. The time reduction increases with larger request volumes. For 1M requests, *Poodle* takes 7.5× less time than the LLM, and for 2M requests more than 10×. Since Llama-2-7B is a small model compared to a flagship LLMs and all requests are wrapped, we expect real-world speedups to be even larger.

**Takeaway:** *For local deployment, even when using a small LLM as the baseline, JITR significantly reduces inference time.*

### 4.3 Accuracy

In this section, we show that small models can compete with LLMs on simple tasks. We use 10,000 random items from the IMDB dataset with a 50/50 training-test split and compare the accuracy of an LLM [29] with the accuracy of a *BERT* [14] model fine-tuned on the ground truth or LLM generated labels. The LLM reaches an accuracy of 0.926 on the training and 0.937 on the test data. The results for the *BERT* model in Table 2 show that a surrogate model reaches competitive accuracy, even though (i) we do not tune hyperparameters, (ii) the LLM might have seen the IMDB data before, (iii) and that *BERT* uses 256 tokens while the LLM sees 1024.

Related work has come to similar conclusions. Across multiple studies and datasets, fine-tuned models (e.g., *RoBERTa*) consistently outperform LLMs on text classification tasks [5, 6, 11, 13]. Pangakis et al. [32] avoid test-set contamination and compare *GPT-4* few-shot performance with small models trained on either human or *GPT-4*–generated labels. Small models trained on the ground-truth slightly outperform *GPT-4*, and those trained on *GPT-4* labels achieve accuracy nearly matching *GPT-4* itself.

**Takeaway:** *For well-scoped tasks, smaller, specialized models achieve accuracy that is competitive with LLMs.*

### 4.4 Model Search

In this section, we show that (1) model search identifies the most promising model and (2) using fine-tuning, outperforms other approaches in development time, accuracy, and required dataset size.

We select ten models from Hugging Face, including the base version of *BERT*, three models trained for sentiment classification [39, 44], and six task-specific non-sentiment classification models to evaluate the rankings produced by model search. We use Alsatian's [40] baseline approach and 500 test and training samples to search the models and generate the ground truth ranking by fully fine-tuning all ten models. Model search ranks domain-specific models lowest, the BERT base model in the middle, and

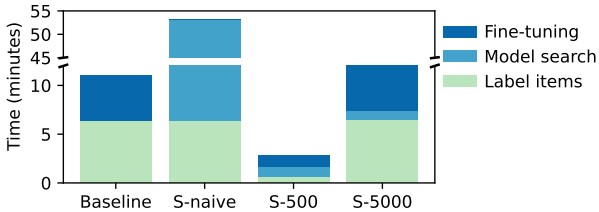

**Figure 6: Time for different model development approaches.**

sentiment classification models highest. The highest-ranked model achieves the highest accuracy when being fully fine-tuned.

We compare four approaches, all utilizing LLM-generated labels, with the goal of reaching 0.89 accuracy. Our baseline collects $n$ LLM-generated labels and then fine-tunes a standard BERT model to evaluate model development time. For this, a minimum of 5,000 items are required to reach the target accuracy of 0.89, which takes eleven minutes. The naive search baseline (*S-naive*) takes 53 minutes to fine-tune all ten available models on 5,000 items and select the best-performing one. It achieves an accuracy of 0.92. The model search approach *S-500* selects the best-performing model and fine-tunes it on 500 training samples. With under three minutes, it completes 4× faster than the baseline and 19× faster than the naive search while reaching an accuracy of 0.91. With 500 samples for model search and 5,000 samples for fine-tuning, *S-5000* takes twelve minutes and matches the accuracy of the naive search baseline while being 4.4× faster. When aiming for the highest accuracy using a naive search approach, model search is the bottleneck. In our experiment, this shifts when using a more advanced search method or more data for fine-tuning than for searching. However, the bottleneck shifts back to model search when searching through all 6,000 fine-tuned BERT variants or all 2M models on Hugging Face. We can mitigate this by a model store that provides system-level optimizations and advanced search capabilities. Following the approach described in Section 2.2, we reduce the effective search space to 70 models. Extrapolating from the *S-5000* setting, which requires approximately one minute to evaluate ten models, this corresponds to an estimated search time of about 7 minutes. By applying additional optimizations, such as partial model access, successive halving [37], or loading compressed models, the overall search overhead can be further reduced, at which point LLM inference for data labeling becomes the dominant bottleneck again.

**Takeaway:** *Model search followed by fine-tuning outperforms alternative model development approaches in development time, accuracy, and required dataset size.*

## 5 SUMMARY AND DISCUSSION

We have demonstrated that JITR significantly reduces inference latency and resource requirements for recurring tasks while providing the same advantages for these tasks as using an LLM: (i) zero manual model development, (ii) no need for manual data collection and labeling, (iii) and no AI expertise required. Given the increasing number of publicly available and privately stored models [12], it is likely that a suitable model exists that can be fine-tuned to produce an accurate surrogate model. However, identifying the right base model among millions of models that will lead to low development cost and good accuracy requires further work on improving the performance and effectiveness of model stores and model search.

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
