# OpenReview forum: "Poodle: Seamlessly Scaling Down Large Language Models with Just-in-Time Model Replacement"
_VLDB.org/2026/Workshop/NOVAS — NOVAS 2026_

### Official Review · Reviewer_PU5R · 2026-07-07

**Confidence:** 4

**Improvement Opportunities:**

a) I understand that this is a vision-ish paper but the empirical analysis beyond cost is a bit weak. For e.g. it is not clear what are the overheads (time, memory, latency etc) for each of the components in the pipeline. Even. a brief discussion would improve the paper.
b) The experimental result is promising in terms of feasibility but also not a bit simple. For example, it mostly focuses on single task/dataset. I worry that sentiment analysis is simple task that might work well. It is not clear how this approach works for more generic tasks. It is also not clear how to define a "task" (ie the task identification problem). In a typical setting, the granularity of JIT optimization is well scoped and understood. It is not clear how to do it here. The authors mention a bunch of ideas but they are closer to heuristics. The paper also raises the issue of drift that makes this even harder.
c) The break even analysis is promising but also a bit simplistic.
d) I wonder how important is model search. Would it be simpler to create some taxonomy and then give predefined models and only do model search within those candidates? For e.g. distilBERT for sentiment. This could also simplify the task problem from previous comment.
e) The human-in/on-the loop is unclear.

**Minor Comments:**

NA

**Short Summary:**

The paper proposes an interesting idea: Just-in-Time Model Replacement (JITR). The idea is to detect recurring LLM-based tasks and transparently replace them with cheaper, task-specific surrogate models. The motivation is to achieve comparable accuracy with  lower cost/latency/energy etc. The authors propose a generic pipeline and build a prototype with promising results.

**Strong Points:**

a) The paper identifies an important and emerging inefficiency of using expensive LLMs for simple but well scoped tasks. The JIT-compilation analogy is an interesting one.
b) I liked the overall pipeline. It is relatively comprehensive and allows lot of customization and opens up many interesting sub-problems.
c) The paper is vision-ish but is backed by actual implementation in GitHub. I had not had a chance to skim the repo. The evaluation is a good start and includes relevant analysis such as break-even points. The analysis is also conservation which is good.
d) The connection to the multitude of related areas of research from model routing, code generation, semantic operators, proxy models is done well.
e) The paper mentions (though briefly) a number of interesting system optimization angles. This elevates the paper from a simple fine-tuning one to one that data management community can produce.

---

### Official Review · Reviewer_SemY · 2026-07-07

**Confidence:** 4

**Improvement Opportunities:**

Thank you for your submission to NOVAS. I enjoyed reading your work. Please find my comments below.

O1. I think the main issue is that this is a quite advanced field already. While the authors make the connection with the related work in the introduction, it is not clear to me why existing work is not sufficient to address this problem. Model-search and model-compression are both well-studied topics and there are a number of inference-serving systems solving a similar problem (see Cocktail, CascadeNet, ModelSwitch, etc.)

O2. Perhaps the use of training data to search for a model is new, yet 36 and 37 seem to have already studied this.

O3. One aspect could be privacy-preserving. Perhaps what the authors propose can be done on the client-side in the API call itself. Different APIs could be called for different use cases.

O4. The problem is new I think -- detecting repetitive tasks. But perhaps existing model search algorithms fail for finding models for repetitive tasks, and this can be a challenge.

O5. For cascading-based approaches, the authors claim that JITR enables replacing proxy with an order-of-magnitude cheaper model. I am not sure what they refer to here... Cascade is a set of models cascaded... It starts from the smallest to the largest...

O6. One thing that is unclear in the study is as follows. The study implies that it uses smaller versions of LLMs (BERT), but its citations 36 and 37 are not necessarily LLMs... They talk about different model architectures, such as ResNet, Inception, etc. So, choosing different models is not the same as choosing a smaller model. The paper should make it clear that they look for different models, even perhaps of the same size. Otherwise, the idea seems to be just choosing a smaller-sized LLM. Then, it is not clear why there should be a model search... Just choose the LLM with a smaller number of parameters... And how much smaller it is depends on your budget.

O7. Section 2.1: Why is it so complicated to detect a repetitive task? Isn't it trivial to identify repeating input-output pairs?

O8. Similarly, isn't it cheap to detect repetitive tasks?

O9. "The most expensive step in feature-based model search is feature extraction (i.e., loading the model to the GPU and performing inference)." --> I am not sure about this. It depends on the base model size and the MLP size... The base model can be a really small one... MLP can be large. And the MLP needs forward and backward passes. So, this statement isn't necessarily true.

O10. "maximize caching of partial models and intermediate results [27, 40, 41]" --> I am not sure what caching refers to here... We cannot cache feature maps across different architectures. Even within the same family of architectures, e.g., ResNet, it might be problematic -- skip connections might not match, etc.

O11. These two dimensions are inter-dependent: (i) minimize the number of models considered for feature-based
search, and (ii) reduce model loading and inference times, if the authors do model pruning. Every pruned model is effectively a new model.

**Minor Comments:**

M1. Abstract's last sentence -- please specify quantitatively how much saving you achieve.
M2. Below Figure 2: based models for fine-tuning --> base models for fine-tuning.

**Short Summary:**

The paper proposes a system that seamlessly integrates smaller-sized models that are specialized for repetitive tasks for LLM inference calls. The problem is important, and the idea is sound, though the paper can benefit from improved argumentation in its comparison with the related work.

**Strong Points:**

S1. Important problem.

S2. Well-written paper.

---

### Official Review · Reviewer_EmML · 2026-07-10

**Confidence:** 4

**Improvement Opportunities:**

O1: Missing related work SEED: Domain-Specific Data Curation With Large Language Models (See ModelGen)
O2: - What is the input to task detection? A series of prompts that are passed to the LLM? How would monitoring the KV-cache help with task detection? Please make this explicit.
O3: What is the goal of minimizing the search space for base model search? Is it to get a diverse set of models? Please make this explicit.
O4: How would caching of partial models help? This part is hard to understand. Please improve the presentation.
O5: The development cost of $4 for finetuning a BERT model seems somewhat arbitrary. What is the inference cost for BERT?
O6: "S-500 selects the best-performing model and fine-tunes it on 500 training samples" --> How is the best-performing model determined? Using Alsatian?
O7: A plot that shows when model search and when fine-tuning are the bottleneck would be helpful to illustrate the last part of 4.4.

**Minor Comments:**

- Figure 2: Please highlight the starting point in the flowchart.
- Figure 3: Hard to understand, what are the small clusterings in the bottom, why are the red lines converging at the top of the arrow?
  --> Maybe switch to a presentation that shows the search procedure, starting from large clusters, and then diving deeper.
- Figure 4: What is the purpose of the Figure? It seems to be already part of Figure 2, and it also takes some time to parse due to the many arrows and abstract icons.
- Figure 6 is never referenced in the text.
- Captions are often non-descriptive.

**Short Summary:**

The authors propose poodle, a system that can replace expensive LLM inference (e.g. in semantic data systems) with automatically fine-tuned models for simple tasks. The authors describe various challenges, such as how to detect tasks automatically, how to generate the dataset for fine-tuning, how to search for a suitable model for fine-tuning, how to fine-tune the model, and finally how to monitor whether the fine-tuned model maintains satisfying latency and accuracy, especially under workload shift. For each aspect, they highlight the open research challenges. In their evaluation, they show that the idea is promising by comparing inference time, monetary cost, accuracy of fine-tuned small LLMs on simple tasks. Finally, they also evaluate several model search approaches and show their importance.

**Strong Points:**

- Well written, mostly easy to read and understand.
- Gives a complete picture of what a JITR system has to consist of, what the design space is, and what challenges there still are.
- Improving latency in semantic data systems is a relevant problem.
- The evaluation shows that the idea is promising.

---

### Decision · Program_Chairs · 2026-07-16

**Decision:**

Accept

**Comment:**

The reviewers agree that Poodle addresses an important problem and presents a clear, comprehensive vision for replacing repetitive LLM inference with cheaper task-specific models. The prototype and evaluation provide promising evidence of cost and latency benefits, and the paper identifies several relevant systems challenges. We recommend  acceptance.